

# Life satisfaction and parental support among secondary school students in Urumqi: the mediation of physical activity

Zehua Zuo[1,*], Shulin Li[2,*], Shuyuan Liu[3] and Qian Wang[1]

[1] School of Public Health, Xinjiang Medical University, Xinjiang, China
[2] Urumqi Municipal Center for Disease Control and Prevention, Xinjiang, China
[3] Urumqi Eye, Ear, Nose and Throat Specialist Hospital, Urumqi, China
[*] These authors contributed equally to this work.

## ABSTRACT

**Background**. This study is aimed to analyze the relationship between parental support, physical activity and life satisfaction among secondary school students in Urumqi city, and also to explore the mediating role of physical activity in the relationship between life satisfaction and parental support among secondary school students.

**Methods**. A questionnaire survey was conducted from December 2021 to January 2022, using a stratified whole-group random sampling method among 2,474 secondary school students aged 11–18 years in Urumqi city, including a general demographic questionnaire, a Chinese version of the Child and Adolescent Physical Activity Questionnaire, Chinese version of the Child and Adolescent Parental Support Scale, and the Multidimensional Student Life Satisfaction Scale. The Bootstrap mediating effect test was used to test the effect relationship between the variables, and Amos was adopted to analyze the mediating effect of physical activity between secondary school students' life satisfaction and parental support.

**Results**. Secondary school students scored $(33.22 \pm 5.68)$ on life satisfaction, $(2.11 \pm 0.64)$ on physical activity, $(24.85 \pm 4.31)$ on the mother's roll and $(24.73 \pm 4.40)$ on the father's roll. Maternal support was a significant direct predictor of life satisfaction $(\beta = 0.303, t = 11.893, P < 0.01)$. Maternal support was a positive predictor of physical activity $(\beta = 0.027, t = 9.332, P < 0.01)$. And the physical activity was a positive predictor of life satisfaction $(\beta = 1.362, t = 7.665, P < 0.01)$. Paternal support was a significant direct predictor of life satisfaction $(\beta = 0.334, t = 13.448, P < 0.01)$. Paternal support was a positive predictor of physical activity $(\beta = 0.030, t = 10.665, P < 0.01)$. Physical activity was a positive predictor of life satisfaction $(\beta = 1.264, t = 7.119, P < 0.01)$. Physical activity had a mediating effect between life satisfaction and parental support (effect values: 11.88% in maternal support and 11.38% in paternal support, respectively).

**Conclusions**. The supportive parental environment of secondary school students in Urumqi can directly influence their life satisfaction and also indirectly influence their life satisfaction by enhancing their physical activity level, and parental support for physical activity of secondary school students should be enhanced to promote their life satisfaction level.

Corresponding author
Qian Wang, 115128726@qq.com, wq115128726@163.com

## INTRODUCTION

As the global phenomenon of population aging increases, the future psychological and social burden on young people will gradually increase and their mental health problems will become more prominent (*Liu & Guo, 2022*). Life satisfaction is the overall perceived assessment of one's life situation over a period of time-based on set criteria (*Diener, Inglehart & Tay, 2013*) and is an essential feature and core of mental health impact (*Yu, 2022*). In recent years, researchers have paid extensive attention to the state of life satisfaction of children and adolescents, with particular focus on the factors influencing their life satisfaction. Studies have shown that life satisfaction is influenced by both internal and external factors, with internal factors including personality, anxiety and optimism, and external factors including social support, life events and interpersonal relationships (*Chen et al., 2017*). Social support can come from family, friends and others (*Ma et al., 2021*) and may affect people's physical and mental health in different ways at different stages of life (*Sheets Jr & Mohr, 2009*). Research suggests that parents or family support are associated with adolescent life satisfaction (*Suldo & Huebner, 2006*). However, research on the relationship between adolescent life satisfaction and family support is still incomplete (*Azpiazu Izaguirre, Fernández & Palacios, 2021*; *Bi et al., 2021*). One of the reasons for the slow progress in research on adolescent life satisfaction may be the lack of consensus on the operationalization of family support structures and conceptual links with other family variables (*Povedano-Diaz, Muñiz Rivas & Vera-Perea, 2020*). With the rapid development of science and technology in society, people's lifestyles are changing and there is a gradual decrease in physical activity (PA) (*Zhang et al., 2017*). Studies have shown that PA has a positive effect on secondary school students' mental health, not only helping to reduce symptoms such as anxiety and depression, but also increasing their life satisfaction and self-esteem levels (*Pierannunzio et al., 2022*; *Lachytova et al., 2017*). *Horodyska et al. (2019)* found that secondary school students' physical activity levels were significantly associated with their home activity environment, even more than the effects of other dimensions on secondary school students' physical activity levels, with parents providing the supportive environment provided by parents, such as encouragement, companionship, and behavioral support, was found to be beneficial in improving children's physical activity levels to some extent (*Kastrati, Gashi & Georgiev, 2020*; *Carver et al., 2021*). Therefore, this study uses a cross-sectional survey to understand the correlation between physical activity, life satisfaction and parental support among secondary school students in Urumqi, and to further explore the mediating effect of physical activity on life satisfaction and parental support, in order to provide a scientific reference for improving the life satisfaction and physical activity levels of secondary school students. This study will provide a scientific reference for improving the life satisfaction of secondary school students and enhancing their physical activity level.

## MATERIALS & METHODS

### Study setting and participants

According to relevant survey results, the prevalence of myopia among primary and secondary school students in Urumqi in 2019 was 36.2% (*Shiet al., 2021*). The sample size estimation equation was as follows: $N = \frac{t^2 \alpha PQ}{d^2}$ ($p$ is the expected prevalence, $d$ is the permissible error, $Q = 1\text{-}p$), set $\alpha = 0.05$, $t_\alpha = 1.96$, permissible error $d = 0.1p$, preliminary estimated sample size $N = 400\,(1 - p)/p = 705$, taking into account factors such as missed interviews and invalid questionnaires, and finally identified around 2,400 students as the study sample. The study was conducted in a cross-sectional survey in Urumqi, Xinjiang Uygur Autonomous Region, Western China, from December 2021 to January 2022. Stratified whole-group random sampling was used to select participants. Two districts (Tianshan District and Shuimogou District) were randomly selected from the seven districts and one county under the jurisdiction of Urumqi city (Xinshi Qu, Tianshan District, Saybag District, Shuimogou District, Toutunhe District, Dabancheng District, Midong District and Urumqi County); five schools (one junior middle school, two senior high schools and two middle schools with integrated junior middle and senior high schools) were then randomly selected and stratified by grade level, and in each eligible school, a proportional random, three classes each from junior middle one to senior high three were sampled and the whole class was sampled as a whole group. A total of 2,506 paper copies of the questionnaires were distributed, and after eliminating invalid questionnaires with missing key information, 2,474 were included in the questionnaire analysis, with a valid return rate of 98.72% and an average student age of (14.31 ± 1.76) years. The inclusion criteria were those who lived in the local area for 12 months, all subjects gave their informed consent and participated voluntarily in the survey. In this study, exclusion criteria were any of the following self-reports: absence from school in the last week for any reason; refusal to cooperate with the questionnaire. The study was approved by the Ethics Review Committee of Urumqi Eye, Ear, Nose and Throat Specialist Hospital (Ethics number: 20211201) and the questionnaire was administered by the students and their legal guardians and signed informed consent forms.

### Measurements

#### General demographic characteristics

A self-administered questionnaire was used to collect general characteristics of the subjects, including sleep duration (<6 h/d, 6–8 h/d, 8–10 h/d, >10 h/d, where the sleep cut-off time was 00:00), parental education (senior high school and below, specialty, undergraduate, postgraduate and above), grade (junior middle 1, junior middle 2, junior middle 3 and senior high 1, senior high 2, senior high 3), total income (<10 million/year, 10–20 million/year, 20–40 million/year, >40 million/year).

### Scale assessment

#### Physical activity level

The Chinese version of The Physical Activity Questionnaire for Older Children and Adolescents (PAQ-CN), prepared by *Kowalski, Crocker & Donen (2004)* and translated

and revised by *Guo (2016)* was used. The scale is a self-administered and retrospective physical activity questionnaire for children and adolescents aged 7-18 years, presenting the distribution of physical activity levels in the child and adolescent population. The Likert four-point questionnaire is designed with nine items: PA1: investigates the frequency of subjects' activity in basketball, dance, *etc.*: (none = 1; 7 times and more = 5); PA2-PA8: assesses the content of subjects' exercise during recess, physical education, outside school, weekends, *etc.* (options 1–5 show the progression of physical activity levels); PA9: investigates subjects' last week (none = 1; very frequent = 5), with each question from PA1 to PA9 being scored from 1 to 5. The scores were summed and averaged to give a physical activity score (retained to two decimal places). The scale has a Cronbach's $\alpha$ of 0.82 and can be used to measure physical activity in Chinese child and adolescent studies.

### Parental support environment

The Chinese version of The Activity Support Scale for Children and Adolescents (ACTS-CN), developed by *Lampard et al. (2016)* and translated and revised by *Guo (2016)* was used. The scale assesses the physical activity support environment of children and adolescents based on students' perceptions of their parents' activity behavior, as well as their parents' attitudes and influence on their own activity behaviors. The scale is divided into a father's and a mother's volume, each consisting of nine items, both using a Likert four-point questionnaire design, which contains two dimensions: Activity Support Behaviors and Attitudes (SS1 to SS6): positive questions (strongly disagree = 1; strongly agree = 4); Activity Behavior Limitations (SS7 to SS9): negative questions (strongly disagree = 1; strongly agree = 4), the number 5 minus the corresponding option is the score for this question, and the total score for the nine questions is the parental support score, with higher scores indicating more parental support. The Cronbach's $\alpha$ for the mother's scale was 0.85 and the Cronbach's $\alpha$ for the father's scale was 0.88, which can be used to assess the parental support of Chinese children and adolescents.

### Life satisfaction status

The Multidimensional Students Life Satisfaction Scale (MSLSS), revised by *Zhang, He & Zheng (2004)* was used. The scale consists of 36 items on a seven-point Likert-type scale (not at all = 1; fully = 7), with questions 3, 4, 9 and 10 being reverse questions. The scale includes six dimensions: friendship satisfaction, environmental satisfaction, school satisfaction, family satisfaction, freedom satisfaction and academic satisfaction, with each dimension being scored as the mean score for the corresponding item. The scale has a Cronbach's $\alpha$ of 0.86 and can be used to assess life satisfaction studies of children and adolescents.

### Quality control

A team of uniformly trained medical students, school health care workers and teachers, *etc.*, will form the investigation team so that using a uniformly printed self-designed questionnaire and standardized measurement scales, a face-to-face survey is conducted according to the program requirements. The researchers explain how to complete the questionnaire according to the criteria and students fill in the questionnaire independently

in the classroom for 15–20 min. While the questionnaire was being completed, the researcher answered the students' questions on the spot and the questionnaire was collected on the spot to check for any logical errors or missing items, *etc.*, which were filled in or corrected by the students on the spot. The questionnaires were double-checked and double-entered.

### Statistical methods

A database was created using EpiData 3.1 and (SPSS) 26.0 (IBM, Armonk, NY, USA) for statistical analysis. Categorical variables were described by number (proportions), and normally distributed continuous variables were described by means and standard deviation (SDs). Comparisons between groups were made by $t$-test or ANOVA, or if normality was not met, median (M) and interquartile spacing ($P_{25}$, $P_{75}$) were used and the Kruskal-Wallis $H$-test was used to make comparisons between groups. The relationship between physical activity, parental support and life satisfaction was examined using Pearson's correlation analysis. Amos 26.0 software (IBM, Armonk, New York, the United States) was used to construct structural equation models, and the mediating effect of physical activity between parental support and life satisfaction was tested using the mediating effect test procedure proposed by *Wen & Ye (2014)*. Bootstrap mediated effects test using the PROCESS version 3.4 plug-in in SPSS 26.0, selecting the bias-corrected non-parametric percentile method, setting $N = 5\,000$, and mediating effects were statistically significant when Bootstrap 95% *CI* did not contain 0. Amos 26.0 software was adopted to construct the mediating effect path analysis graph. The level of significance was set at 0.05, two-tailed.

## RESULTS

### Basic information

Table 1 shows the characteristics of all included participants, among the 2,474 secondary school students surveyed in Urumqi, boys (45.8%) and girls (54.2%). Junior middle 1 students (29.2%), junior middle 2 students (26.2%), junior middle 3 students (8.6%), senior high 1 students (12.0%), senior high 2 students (16.9%) and senior high 3 students (7.2%). It was noted that 1.6% of mothers and 2.1% of fathers had completed postgraduate studies or above. A total of 2.8% of participants recalled a total household income greater than RMB 400,000 per year. A total of 1,522 participants (61.5%) reported sleeping 6–8 h per day.

### Life satisfaction, physical activity, and parental support

The total mean score of life satisfaction among secondary school students in Urumqi was (193.77 ± 33.28), physical activity score (2.11 ± 0.64), maternal support score (24.85 ± 4.31), and paternal support score (24.73 ± 4.40). The differences in life satisfaction were statistically significant ($P < 0.05$) across the school year, gender, sleep duration, parental education, and total income. The differences in physical activity were statistically significant across the school year, gender, sleep duration, and total income ($P < 0.05$). The differences in maternal support were statistically significant across school years, gender, hours of sleep, and parental education ($P < 0.05$). The differences in paternal support

**Table 1 Comparison of life satisfaction, physical activity and parental support scores of secondary school students in Urumqi on demographic variables.**

| Variable | Sample | Life satisfaction | PA | SS_Mo | SS_Fa |
|---|---|---|---|---|---|
| Grade level | | | | | |
| Junior middle 1 | 723 (29.2) | 34.3 (30.9, 37.3) | 2.2 (1.8, 2.6) | 25.66 ± 4.29 | 25.60 ± 4.52 |
| Junior middle 2 | 647 (26.2) | 34.2 (29.5, 37.8) | 2.2 (1.7, 2.7) | 24.77 ± 4.41 | 24.62 ± 4.36 |
| Junior middle 3 | 213 (8.6) | 34.9 (30.4, 37.5) | 2.1 (1.7, 2.6) | 25.62 ± 4.16 | 25.16 ± 4.38 |
| Senior high1 | 296 (12.0) | 33.0 (29.0, 36.2) | 1.7 (1.4, 2.1) | 24.36 ± 3.77 | 24.21 ± 3.89 |
| Senior high 2 | 417 (16.9) | 33.4 (29.6, 37.1) | 1.7 (1.4, 2.1) | 23.91 ± 4.19 | 23.72 ± 4.29 |
| Senior high 3 | 178 (7.2) | 31.8 (28.3, 34.9) | 1.8 (1.6, 2.2) | 23.98 ± 4.52 | 24.23 ± 4.48 |
| *F/H* | | 36.745 | 248.205 | 13.113 | 12.141 |
| *P value* | | <0.001 | <0.001 | <0.001 | <0.001 |
| Gender | | | | | |
| Boys | 1134 (45.8) | 33.80 ± 5.72 | 2.24 ± 0.68 | 24.44 ± 4.26 | 24.58 ± 4.32 |
| Girls | 1340 (54.2) | 32.73 ± 5.61 | 2.00 ± 0.57 | 25.20 ± 4.32 | 24.85 ± 4.47 |
| *t/H* | | 4.664 | 9.838 | −4.417 | −1.534 |
| *P value* | | <0.001 | <0.001 | <0.001 | 0.125 |
| Sleep (h/d) | | | | | |
| <6 | 491 (19.9) | 32.0 (27.5, 36.0) | 1.8 (1.4, 2.2) | 24.0 (21.0, 27.0) | 24.0 (21.0, 27.0) |
| 6–8 | 1522 (61.5) | 33.8 (30.0, 37.1) | 2.0 (1.6, 2.5) | 25.0 (23.0, 27.0) | 25.0 (22.0, 27.0) |
| 8–10 | 442 (17.9) | 35.4 (31.5, 38.5) | 2.3 (1.9, 2.8) | 26.0 (23.0, 27.0) | 26.0 (23.0, 28.0) |
| >10 | 19 (0.8) | 38.3 (29.5, 41.7) | 2.3 (1.6, 3.3) | 26.0 (19.0, 28.0) | 27.0 (20.0, 28.0) |
| *F/H* | | 79.342 | 143.348 | 32.306 | 39.544 |
| *P value* | | <0.001 | <0.001 | <0.001 | <0.001 |
| Mo_edu | | | | | |
| Senior high school and blow | 1325 (53.6) | 32.88 ± 5.63 | 2.0 (1.6, 2.5) | 24.61 ± 4.26 | 24.52 ± 4.35 |
| Specialties | 647 (26.2) | 33.56 ± 5.81 | 2.0 (1.6, 2.5) | 24.75 ± 4.46 | 24.60 ± 4.45 |
| Undergraduate | 463 (18.7) | 33.64 ± 5.56 | 2.0 (1.6, 2.5) | 25.63 ± 4.16 | 25.33 ± 4.45 |
| Postgraduate and above | 39 (1.6) | 34.31 ± 6.01 | 2.1 (1.6, 2.9) | 25.67 ± 3.96 | 26.79 ± 3.74 |
| *F/H* | | 3.708 | 2.598 | 7.032 | 7.002 |
| *P value* | | 0.011 | 0.458 | <0.001 | <0.001 |
| Fa_edu | | | | | |
| Senior high school and blow | 1368 (55.3) | 32.91 ± 5.63 | 2.0 (1.6, 2.5) | 24.62 ± 4.26 | 24.45 ± 4.40 |
| Specialties | 628 (25.4) | 33.74 ± 5.85 | 2.0 (1.7, 2.5) | 24.95 ± 4.38 | 24.71 ± 4.30 |
| Undergraduate | 427 (17.3) | 33.38 ± 5.58 | 2.0 (1.6, 2.4) | 25.33 ± 4.28 | 25.37 ± 4.45 |
| Postgraduate and above | 51 (2.1) | 33.92 ± 5.51 | 2.3 (1.7, 3.1) | 25.88 ± 4.39 | 26.71 ± 4.23 |
| *F/H* | | 3.492 | 7.212 | 4.114 | 8.352 |
| *P value* | | 0.015 | 0.065 | 0.006 | <0.001 |

*(continued on next page)*

**Table 1** (*continued*)

| Variable | Sample | Life satisfaction | PA | SS_Mo | SS_Fa |
|---|---|---|---|---|---|
| Total income (million/year) | | | | | |
| <10 | 916 (37.0) | 32.60 ± 5.80 | 2.06 ± 0.61 | 24.66 ± 4.43 | 24.48 ± 4.37 |
| 10–20 | 1326 (53.6) | 33.63 ± 5.46 | 2.12 ± 0.64 | 25.01 ± 4.17 | 24.86 ± 4.37 |
| 20–40 | 162 (6.5) | 33.38 ± 6.15 | 2.15 ± 0.65 | 24.88 ± 4.50 | 25.07 ± 4.69 |
| >40 | 70 (2.8) | 33.25 ± 6.39 | 2.32 ± 0.69 | 24.29 ± 4.82 | 24.69 ± 4.78 |
| *F/H* | | 5.928 | 4.733 | 1.595 | 1.670 |
| *P value* | | <0.001 | 0.003 | 0.189 | 0.171 |

**Notes.**

PA, Physical Activity Score; SS_Fa, Parental Support of Father; SS_Mo, Parental Support of Mother; Fa_edu, Father's education; Mo_edu, Mother's education.

were statistically significant in terms of different school years, sleep duration and parental education ($P < 0.01$) (Table 1).

## Correlation analysis of physical activity, parental support and life satisfaction

Pearson's correlation analysis showed that life satisfaction scores were positively correlated with physical activity scores ($r = 0.223$, $P < 0.01$), physical activity scores were positively correlated with maternal support and paternal support ($r = 0.201, 0.230$, $P < 0.01$), and life satisfaction scores were positively correlated with maternal support and paternal support were also positively correlated ($r = 0.246, 0.277$, $P < 0.01$) (Table 2).

## The relationship between parental support, physical activity and life satisfaction

### Relationship between maternal support, physical activity and life satisfaction

Model 4 tests using PROCESS showed that maternal support was a significant direct predictor of life satisfaction after controlling for sleep duration ($\beta = 0.303$, $t = 11.893$, $P < 0.01$). Maternal support had a positive predictive effect on physical activity ($\beta = 0.027$, $t = 9.332$, $P < 0.01$). And physical activity had a positive predictive effect on life satisfaction ($\beta = 1.362$, $t = 7.665$, $P < 0.01$) (Table 3).

### Relationship between paternal support, physical activity and life satisfaction

The results of the Model 4 test using PROCESS showed that paternal support was a significant direct predictor of life satisfaction after controlling for sleep duration ($\beta = 0.334$, $t = 13.448$, $P < 0.01$), paternal support was a positive predictor of physical activity ($\beta = 0.030$, $t = 10.665$, $P < 0.01$), and physical activity was a positive predictor of life satisfaction ($\beta = 1.264$, $t = 7.119$, $P < 0.01$) (Table 4).

## Mediating effects of parental support in the relationship between physical activity and life satisfaction

### Mediating effect of maternal support between physical activity and life satisfaction

With maternal support as the independent variable and life satisfaction as the dependent variable, the results of the structural equation model using Amos showed that the direct

**Table 2  Correlation analysis of life satisfaction, physical activity and parental support among secondary school students in Urumqi.**

| Variables | 1 | 2 | 3 | 4 |
|---|---|---|---|---|
| 1. Life satisfaction | 1.000 | | | |
| 2. SS_Fa | 0.277[*] | 1.000 | | |
| 3. SS_Mo | 0.246[*] | 0.717[*] | 1.000 | |
| 4. PA | 0.223[*] | 0.230[*] | 0.201[*] | 1.000 |

Notes.
  *P value < 0.001.
  PA, Physical Activity Score; SS_Fa, Parental Support of Father; SS_Mo, Parental Support of Mother.

**Table 3  Test of mediating effects of physical activity (mother's version) ($n = 2,474$).**

| Regression equation | | Fit the indicator | | | Coefficient significance | |
|---|---|---|---|---|---|---|
| Result variables | Predictors | $R$ | $R^2$ | $F$ | $\beta$ | $t$ |
| Life satisfaction | | 0.293 | 0.086 | 116.143 | | |
| | SS_Mo | | | | 0.303 | 11.893[*] |
| | Sleep | | | | 1.430 | 8.313[*] |
| PA | | 0.292 | 0.085 | 114.980 | | |
| | SS_Mo | | | | 0.027 | 9.332[*] |
| | Sleep | | | | 0.211 | 10.977[*] |
| Life satisfaction | | 0.327 | 0.107 | 98.820 | | |
| | SS_Mo | | | | 0.267 | 10.411[*] |
| | Sleep | | | | 1.142 | 6.559[*] |
| | PA | | | | 1.362 | 7.665[*] |

Notes.
  *P value < 0.001.
  PA, Physical Activity Score; SS_Mo, Parental Support of Mother.
  All variables in the model are brought into the regression equation using standardized variables.

effect of Bootstrap 95% *CI* of physical activity as a mediating variable between life satisfaction and maternal support was (0.025 to 0.050), excluding 0, indicating that physical activity has a mediating role, the mediating effect(a*b) = total effect(c)-direct effect(c') =0.036, and the mediating effect model is shown in Fig. 1, where physical activity as a mediating effect accounts for 11.88% of the total effect and the direct effect is 88.12% of the total effect (Table 5).

### Mediating effect of paternal support between physical activity and life satisfaction

Using paternal support as the independent variable and life satisfaction as the dependent variable, the results of structural equation modeling using Amos showed that the Bootstrap 95% *CI* for the direct effect of physical activity as a mediating variable between life satisfaction and paternal support was (0.025 to 0.051), excluding 0, then indicating that physical activity mediated the effect, the mediating effect(a*b) = total effect(c)-direct effect(c') = 0.038 and the mediating effect model is shown in Fig. 2, where physical activity

**Table 4** Test of mediating effects of physical activity (father's version) ($n = 2,474$).

| Regression equation | | Fit the indicator | | | Coefficient significance | |
|---|---|---|---|---|---|---|
| Result variables | Predictors | $R$ | $R^2$ | $F$ | $\beta$ | $t$ |
| Life satisfaction | | 0.315 | 0.100 | 136.522 | | |
| | SS_Fa | | | | 0.334 | 13.448[*] |
| | Sleep | | | | 1.357 | 7.934[*] |
| PA | | 0.308 | 0.095 | 129.060 | | |
| | SS_Fa | | | | 0.030 | 10.665[*] |
| | Sleep | | | | 0.205 | 10.661[*] |
| Life satisfaction | | 0.343 | 0.118 | 109.740 | | |
| | SS_Fa | | | | 0.297 | 11.787[*] |
| | Sleep | | | | 1.099 | 6.342[*] |
| | PA | | | | 1.264 | 7.119[*] |

**Notes.**
[*]$P$ value $< 0.001$.
PA, Physical Activity Score; SS_Fa, Parental Support of Father.
All variables in the model are brought into the regression equation using standardized variables.

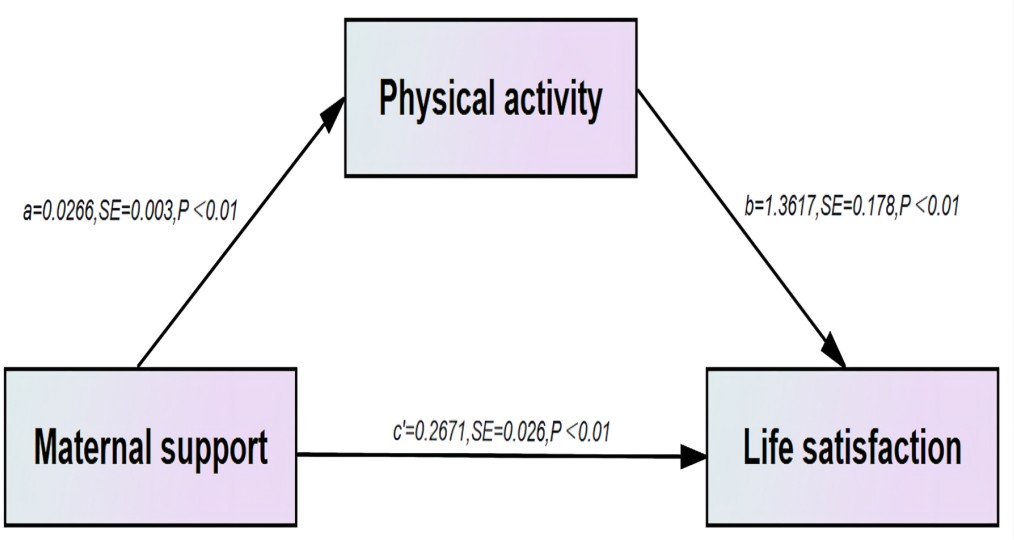

**Figure 1** Analytical model of the mediating role of physical activity in the association between maternal support and life satisfaction. .

as a mediating effect accounted for 11.38% of the total effect and the direct effect accounted for 88.62% of the total effect (Table 6).

# DISCUSSION

The results of this cross-sectional study showed that the total life satisfaction score of secondary school students in Urumqi ($193.77 \pm 33.28$) was higher than the life satisfaction score ($165.67 \pm 28.84$) in the study of *Xue (2020)* on secondary school students in Lvliang,

**Table 5** Bootstrap analysis of the mediating effect of physical activity between maternal support and life satisfaction.

| Effect type | Path | Effect value | Boot SE | Boot 95% CI | Amount of effect (%) |
|---|---|---|---|---|---|
| Mediation effect | SS_Mo →PA →Life satisfaction | 0.036 | 0.006 | 0.025∼0.050 | 11.88 |
| Direct effect | SS_Mo →Life satisfaction | 0.267 | 0.026 | 0.217∼0.317 | 88.12 |
| Total effect | | 0.303 | 0.026 | 0.253∼0.353 | |

**Notes.**
PA, Physical Activity Score; SS_Mo, Parental Support of Mother.

**Table 6** Bootstrap analysis of the mediating effect of physical activity in the relationship between paternal support and life satisfaction.

| Effect type | Path | Effect value | Boot SE | Boot 95% CI | Amount of effect (%) |
|---|---|---|---|---|---|
| Mediation effect | SS_Fa →PA →Life satisfaction | 0.038 | 0.007 | 0.025∼0.051 | 11.38 |
| Direct effect | SS_Fa →Life satisfaction | 0.296 | 0.025 | 0.247∼0.346 | 88.62 |
| Total effect | | 0.334 | 0.025 | 0.285∼0.383 | |

**Notes.**
PA, Physical Activity Score; SS_Fa, Parental Support of Father.

Shijiazhuang, Wuhu, and Fuzhou, and the life satisfaction score (162.21 ± 30.55) of secondary school students in Shangqiu in the study of *Li (2021)*, and the life satisfaction score (181.37 ± 30.13) in the study of *Qiao, Sun & Liu (2021)* on students aged 11-18 in Siping, Baicheng, Jinan, Jilin, suggesting a relatively good life satisfaction situation for secondary school students in Urumqi. In this study, boys had higher life satisfaction scores (33.80 ± 5.72) than girls (32.73 ± 5.61), and the results suggest that there is a relationship between life satisfaction and gender among secondary school students, which may be related to the psychological, social and physical changes that occur during adolescence (physical and hormonal changes, thoughts, social relationships, roles *etc.*), or may be due to gender-specific characteristics (boys tend to experience negative emotions more often than girls, are better able to manage their emotions and are more physically active) (*Azpiazu Izaguirre, Fernández & Palacios, 2021*; *Lyons et al., 2013*). In particular, high school students scored lower in life satisfaction than junior high school students, possibly because high school students face greater exam pressure and are more likely to feel dissatisfied with their lives and school environment due to increased academic and psychological stress (*Xu et al., 2016*). The physical activity score of secondary school students in Urumqi was (2.11 ± 0.64), which was lower than the physical activity score of (2.71 ± 0.83) in the study conducted by *Zhang (2018)* on secondary school students in Nanjing, which may be related to the decrease in physical activity due to the gradual increase in academic stress and inconsistent emphasis on exercise as the school year progresses. Maternal support, paternal support scores were (24.85 ± 4.31) and (24.73 ± 4.40), respectively were higher than the parental support scores of (20.80 ± 5.45) and (17.44 ± 8.87) in the Nanchang study by *Wang (2020)*, suggesting a relatively better environment for parental support for secondary school students in Urumqi.

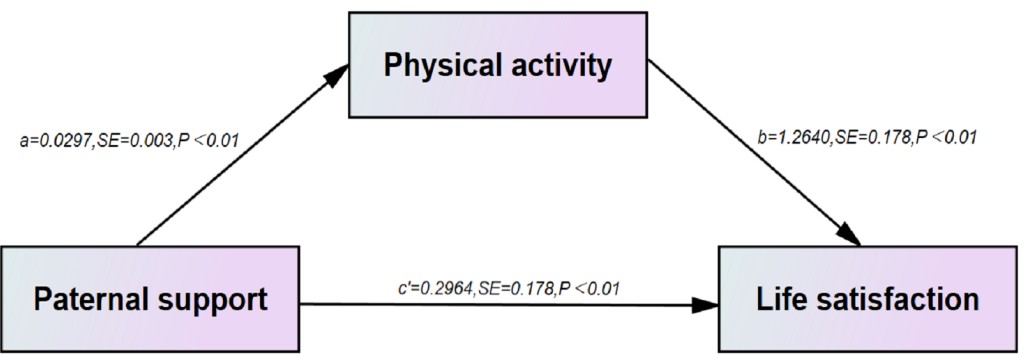

**Figure 2  Analytical model of the mediating role of physical activity in the association between paternal support and life satisfaction.**

## Relationship between physical activity, parental support and life satisfaction

This study found that physical activity was positively associated with parental support, consistent with the study by *Tu, Watts & Masse (2015)*. Parental support for physical activity in secondary school students conveys a positive attitude towards health, life and physical activity, can motivate students to develop a positive physical activity state and achieve better self-development (*Xu & Dong, 2021*), which is in line with the findings of *Shi & Wang (2021)*, which may be due to the fact that secondary school students in adolescence are prone to anxiety due to academic pressure, *etc.*, and when they receive parental support, it is beneficial to adopt a positive coping style and thus increase life satisfaction. This may be due to the fact that adolescent students are prone to anxiety due to academic stress, and when they have parental support, it is beneficial for them to adopt positive coping styles and thus increase their life satisfaction. The results of this study showed that physical activity was positively associated with life satisfaction among secondary school students. *Xu et al. (2018)* found that moderate-to-high intensity physical activity had a positive effect on life satisfaction (*Greenwood, 2019*). Secondary school students should be encouraged to engage in more physical activities to improve their life satisfaction. Among them, girls scored lower in life satisfaction than boys in terms of physical activity level.

## Physical activity has a mediating effect between life satisfaction and parental support

The results in this study showed a mediating effect of physical activity between life satisfaction and parental support, indicating that parental support holding can affect secondary school students' life satisfaction not only directly but also indirectly through physical activity, suggesting a role for physical activity between parental support and life satisfaction. However, the mediating effect values were relatively small, in which the mediating effect values between physical activity and life satisfaction and maternal support and paternal support were 11.88% and 11.38% respectively, and the differences were not statistically significant, which may be related to the small sample size and the small scope of the study area in this study, and the sample size and study area can be further

expanded in the follow-up study. Among them, the effect of a supportive environment for fathers on secondary school students' physical activity was slightly greater than that of mother's supportive environment, which is consistent with the study by *Liu et al. (2022)*. Related research suggests that fathers play a key role in promoting their children's physical activity and influencing their choices and behaviors (*Zahra, Sebire & Jago, 2015*) and that having fathers participate in physical activities with their children can promote increased physical activity in children (*Morgan et al., 2014*). The results of this study show that parental support environments have different effects on secondary school students' life satisfaction, with fathers' support having a greater impact on secondary school students' life satisfaction. Some studies have shown that parenting style is a key factor influencing the level of physical activity participation of adolescents (*Wang, 2019*), while parenting style largely influences the healthy growth of an individual's personality and level of psychological well-being (*Wang, Yang & Shan, 2022*), both parents have different parenting styles, with fathers being higher than mothers on the rejection dimension and the opposite on the emotional warmth and overprotection dimension (*Zhang, Li & Ku, 2022*), which may be due to the "strict father and loving mother" dimension in China. This may be due to the gender roles of "strict father and loving mother" in China. Liu's study found that fathers have a significant impact on children's self-efficacy (*Liu, 2020*), suggesting that paternal support can alleviate fathers' anxiety from mother to child (*Liu, Guo & Chen, 2022*), resulting in higher life satisfaction among secondary school students when fathers are supportive.

In summary, both parents, especially fathers, should encourage appropriate physical activity in secondary school students, increase support for physical activity in secondary school students, and enhance secondary school students' perceptions of their life state. Secondary school students should take an active role in physical activity or outdoor sports, develop a wide range of hobbies and interests, and maintain a positive attitude towards life to further enhance life satisfaction.

Although this study has some important findings, some limitations should be acknowledged. First, as this study adopts a local area cross-sectional survey, it can only show that there is an association between parental support, physical activity and life satisfaction, and cannot make inferences about the causal relationship between life satisfaction and physical activity and parental support, which can be further explored using cohort studies. Second, there may be information bias (or recall bias) in the secondary school students' recall of parental educational level during the process of completing the questionnaire. In addition, the study sample was limited to one area of Urumqi city, and there may be measurement and sampling errors in the survey process, and the objectivity of the results needs to be further verified. Finally, the lack of information on Socio-Economic Status (SES) in this paper may cause some errors in the results of this study, which should be further explored in later studies. The findings in this study have the following practical implications: on the one hand, schools, parents and society should recognize the importance of parental support and physical activity levels in improving the life satisfaction of secondary school students and suggest that parents, especially fathers, should encourage their children to be physically active appropriately and increase their support for physical

activity. On the other hand, the importance of physical activity in promoting the mental health of secondary school students is emphasized. Secondary school students should engage in appropriate physical activities in their own lives in order to develop positive emotions, which also contribute to their life satisfaction.

## CONCLUSIONS

(1)  After controlling for sleep duration, parental support was a significant and positive predictor of life satisfaction among secondary school students.

(2)  Parental support was not only a significant direct predictor of life satisfaction for secondary school students, but also an indirect predictor of life satisfaction through the mediating role of physical activity.

(3)  In this study, the findings may have expanded the understanding of the relationship between life satisfaction and parental support among secondary school students and focused on interventions to encourage appropriate physical activity and enhance life satisfaction among secondary school students.

## ACKNOWLEDGEMENTS

The authors thank all participants and investigators.

### Funding

This work was supported by the Natural Science Foundation of Xinjiang Uygur Autonomous Region (2017D01C186) and the Public Health and Preventive Medicine, a special discipline of the 14th Five-Year Plan of Xinjiang Uygur Autonomous Region. The funders had no role in study design, data collection and analysis, decision to publish, or preparation of the manuscript.

### Grant Disclosures

The following grant information was disclosed by the authors:
Natural Science Foundation of Xinjiang Uygur Autonomous Region: 2017D01C186.
Public Health and Preventive Medicine, a special discipline of the 14th Five-Year Plan of Xinjiang Uygur Autonomous Region.

### Competing Interests

The authors declare there are no competing interests.

### Author Contributions

- Zehua Zuo conceived and designed the experiments, performed the experiments, analyzed the data, prepared figures and/or tables, authored or reviewed drafts of the article, and approved the final draft.
- Shulin Li conceived and designed the experiments, performed the experiments, authored or reviewed drafts of the article, and approved the final draft.

- Shuyuan Liu conceived and designed the experiments, authored or reviewed drafts of the article, and approved the final draft.
- Qian Wang conceived and designed the experiments, authored or reviewed drafts of the article, and approved the final draft.

## Human Ethics

The following information was supplied relating to ethical approvals (*i.e.*, approving body and any reference numbers):

The study was approved by the Ethics Committee of Urumqi Eye, Ear, Nose and Throat Specialist Hospital (no. 20211201).

## Data Availability

The raw data is available as in the Supplemental File.

## Supplemental Information

Supplemental information for this article can be found online at http://dx.doi.org/10.7717/peerj.14122#supplemental-information.

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
