# Peer review of "Life satisfaction and parental support among secondary school students in Urumqi: the mediation of physical activity"

_PeerJ, doi:10.7717/peerj.14122_

## Round 0.1 · original submission · Major Revisions

Dear authors,

Respond point by point and address the reviewers' comments. The manuscript needs a major revision.

Reviewer 1 ·

Basic reporting

• Line 15, please check bold word “so as to provide ascientific reference…”
• Line 46, please add “dot” after “providing”;
• Line 64, please check the period: “The requirements of [9]…”
• Line 70, please add “dot” after “january 2022”;
• Line 75, there's a space missing between the two words;
• Line 88, complete the period.
• Line 102 and line 108, please check the periods.
• Line 200, there's a space missing between the two words;

Experimental design

Your methods needs more detail. I suggest that you to provide more justification for the following points:
- The target popultaion is young people, so is there parent’s consent form? Please specify this point and if there is not parent’s consent specify “why”.
- Please specify the instruments of data collection: are the paper or online questionnaire? How are administred this questionnaire (in class, at home)?
- Please specify the teachers and local staff role for this data collection.
- The schools can refuse partecipation to the study? Please specify.
- Please specify the sleep cutoff which you are adopted for the analysis.
-Please add more detail about the 2 urban area selected for the sample (for example population density, etc.)

Validity of the findings

• Your results are compelling, the data analysis should be improved in the following ways:
- we suggest to add the mean age for each school’s grade of the population enrolled;
- are there differences among the 2 urban area selected for the analysis?
- Are there information about socio economic status (SES)? This is an important determinant of outcome.

Additional comments

• I think it would be better to add some points about the limitation of the study; in particular:
- The young people completed the questionnaire so they respond about parental educational level; this procedure can introduce an information bias (or recall bias). I suggest to add this limitation in the discussion.
- If there are not information about SES, i suggest to add this limitation in the discussion of results.

Reviewer 2 ·

Basic reporting

Thank you for the opportunity to review this manuscript. This is an interesting study which has investigated the mediation effect of physical activity in the effect of parental support on life satisfaction. However, the manuscript has several shortcomings, in particular:

1) The introduction and context is not well referenced and relevant, considering the importance of the topic and the ample evidence
2) The tables are not clear in the description and difficult to understand

Experimental design

The survey through the administration of the questionnaires is not very clear and the methods described provide limited indications, especially related to physical activity

Validity of the findings

The methods described require further investigation with more detailed conclusions

·

Basic reporting

The manuscript describes a research that deals with a topic of great interest but it is weak in its methodological structure.

Experimental design

The research shows gaps in background and sample recruitment. It could be better described how the champion was recruited.

Validity of the findings

The conclusions are also unconvincing. The perspectives and limitations of the research also need to be better described. The contribution to the existing literature should be specified.

---

## Round 0.2 · accepted · Accept

Dear authors,
congratulations!
You have made the revisions requested by the reviewers and the manuscript is much improved.